# Comparison of Three Different IBD Vaccination Protocols in Broiler Chicken—Efficacy, Serological Baselines and Histo-Pathological Lesions in the Bursa of Fabricius

**DOI:** 10.3390/ani14213116

**Published:** 2024-10-29

**Authors:** Marcin Śmiałek, Joanna Kowalczyk, Michał Gesek

**Affiliations:** 1Department of Poultry Disease, Faculty of Veterinary Medicine, University of Warmia and Mazury, 10-719 Olsztyn, Poland; joanna.kowalczyk@uwm.edu.pl; 2SLW Biolab Veterinary Laboratory, ul. Grunwaldzka 62, 14-100 Ostróda, Poland; 3Department of Pathological Anatomy, Faculty of Veterinary Medicine, University of Warmia and Mazury, 10-719 Olsztyn, Poland; michal.gesek@uwm.edu.pl

**Keywords:** Gumboro disease, vaccination, live vaccine, ELISA serological baselines

## Abstract

Infectious bursal disease (IBD) is a highly infectious disease of chickens found in all latitudes, due to the very high resistance to environmental conditions and commonly used disinfectants of the IBD virus (IBDV). IBD is a disease of high economic impact on the broiler industry. Various vaccines and vaccination strategies have been developed to immunize broilers against IBD. Milder vaccines are used when there is less pressure from field viruses, while more immunogenic vaccines are used when there is high pressure from IBDV. The more immunogenic vaccines also have a higher probability of damaging the structures of the birds’ immune systems. This study was conducted to compare three strategies for immunizing broilers against IBD, differing in the degree of invasiveness of the vaccine viruses. The results showed unequivocal differences in the levels of immunogenicity of the different vaccination programs. These differences were also directly related to the degree of damage to the structures of the immune system after vaccination. The obtained results allow the adequate selection of a given IBD vaccination program, depending on the current epidemiological situation of IBDV, while maintaining a minimal level of damage to the structures of the immune system of the birds.

## 1. Introduction

Infectious bursal disease (IBD), also known as Gumboro disease, is highly contagious and highly infectious and one of the most economically significant diseases in chicken. The etiological agents of IBD are viruses possessing double-stranded RNA belonging to the *Birnaviridae* family [1,2,3,4].

The greatest susceptibility to the onset of clinical IBD is demonstrated by chickens between 3 and 6 weeks of age, which correlates with the dynamic development of the bursa of Fabricius (BF) that takes place in birds in this particular age range. This susceptibility is due to the fact that the target cells for IBDV are B cells maturing in the stroma of the BF. The result of IBD in chicken is immunosuppression as a consequence of damage to B cells which are responsible for the induction of humoral immunity of the macroorganism [4,5,6,7,8].

IBD viruses are found in all latitudes due to their very high resistance to environmental conditions and commonly used disinfectants. It has been shown that after the removal of diseased birds and manure, a poultry house contaminated with IBD virus poses a risk of infection for up to 4 months [9,10]. An additional factor highlighting the problematic nature of IBD is the specific immunoprophylaxis of this disease in young birds. Vaccination is a constant balancing act between “too early” and “too late” administration of the vaccine virus, which is caused by high and/or diversified levels of maternal antibodies in the first weeks after chicks hatch [4,11,12].

Several schemes for specific immunoprophylaxis of IBD have been developed, and one of the most widely used schemes is the administration of live attenuated vaccines with drinking water on the optimal day—which should be equated with the moment when the level of maternal antibodies falls to a level that can be breached by the attenuated vaccine virus. The determination of this optimal date is based on the principles described in the Deventer formula [2,3,12,13].

The Deventer formula, which uses conventional ELISA tests and compatible computer software for them, is a tool designed to allow the determination of the optimal timing for IBD vaccination. The parameter assessed in this formula that enable this calculation is the baseline maternal antibody level measured in the serum of chicks in the first days after hatching (with ELISA tests). Knowing the level of anti-IBDV antibodies in the first days of life and knowing their half-life periods (3 days for chicken broilers) allow us to calculate when the level of these antibodies falls to the level which can be overcome by the specific vaccine virus [2,14]. Live attenuated vaccines against IBD have strictly defined antibody titers that they are able to break through. These are most generally divided (based on the level of attenuation) into mild vaccines that break through maternally derived antibodies (MDAs) titers of 125, intermediate vaccines that break through MDAs titers of 250 and intermediate plus vaccines that break through MDAs titers of 500. Finally, hot vaccines break through titers of more than 500 [3,4].

The choice of a given vaccine and the timing of its use in a given flock is most often dictated by several factors, the most important of which are the environmental IBD virus pressure and the level and uniformity of the maternal antibodies measured by a coefficient of variation (CV). In some cases, when the CV of the MDAs is above 40%, specific rules of IBD vaccination might be applied in the flock [2,14].

Another problematic element of IBD vaccination programs is the evaluation of their safety and efficacy. While efficacy can be expressed in terms of the degree of stimulation of the immune system, estimated on the basis of the levels of antibodies found at the end of the production cycle of chicken broilers, the safety of their use should be based on an assessment of the degree of histopathological changes in the BF. 

Taking the above into account, this study was conducted to evaluate three different IBD vaccination protocols in broiler chicken to measure their effectiveness; an estimation of serological baseline using two different ELISA tests and the degree of progression of BF lesions after vaccination were assessed. This study was designed to reflect as much as possible the condition of the birds under field conditions after IBD vaccination. This study was performed on industrial farms, and the vaccination protocols assumed the commonly practiced re-vaccination of birds against IBD. To our best knowledge, this is the first scientific work of its kind on IBD vaccination.

## 2. Materials and Methods

### 2.1. Experimental Layout

This experiment was conducted under field conditions on commercial broiler farms. Prior to the experiment, the farms were evaluated for compliance with animal welfare requirements and good husbandry practices. A total of 42 farms were tested, including 13 vaccinated with protocol I, 15 vaccinated with protocol II and 14 vaccinated with protocol III (see below). These farms were under the veterinary service of six independent veterinary practices. Vaccination with Avishield IBD INT and/or Avishield IBD PLUS vaccines (Dechra Veterinary Products, Poland) was performed with drinking water at optimal vaccination times (calculated based on the Deventer formula) in accordance with generally accepted good veterinary practices. An adequate number of doses (reflecting the number of birds in the flock) of the appropriate vaccine was dissolved in water and administered with drinking water via a medicated dispenser, so that the vaccine solution was drunk by the birds within a maximum of 2–3 h. Within the experiment, three different vaccination protocols were applied:Single vaccination using an intermediate plus vaccine (MDAs breakthrough: 500; Avishield IBD PLUS). The vaccination timing was estimated based on the Deventer formula, with the assumption of the optimal time of vaccination for 75% of the flock.Double vaccination with an intermediate vaccine (MDAs breakthrough: 236; Avishield IBD INT) [15]. The timing of the first vaccination was estimated based on the Deventer formula, with the assumption of the optimal time of vaccination for 50% of the flock, with re-vaccination 5 days later.Double vaccination with an intermediate plus vaccine (MDAs breakthrough: 500; Avishield IBD PLUS). The timing of the first vaccination was estimated based on the Deventer formula, with the assumption of the optimal time of vaccination for 50% of the flock, with re-vaccination 5 days later.

Blood samples were collected from the birds on the first day of their lives in order to apply the Deventer formula. For vaccination protocols II and III, sample collection of the bursa of Fabricius (BF) for histopathological examination (HP) and histopathological lesion score (HLS) evaluation were collected on the day of the second vaccination. At approximately six weeks of age, BF samples for HP and HLS, BF samples for PCR testing (to differentiate between virulent and non-virulent IBDV) and blood samples for serology were collected from birds on farms vaccinated under all three protocols. The number of samples collected per flock at each time point was 23 for blood and 5 for BF. Production results were made available from all the test farms. Only farms vaccinated with protocols I and II had one case each of field vvIBDV infection (results from these farms were excluded from the calculation of serological baselines).

### 2.2. Serology

Infectious bursal disease serological evaluations in six-week-old birds were performed with two commercial ELISA kits (IDEXX, Westbrook, ME, USA and BioChek, Reeuwijk, The Netherlands). Only the IDEXX IBD ELISA kit was used to determine the level of MDAs and to establish the vaccination timing based on the Deventer formula. Successive steps of the ELISA tests were performed according to the manufacturer’s recommendations. ELISA was carried out with the use of the Eppendorf epMotion 5075 LH automated pipetting station (Eppendorf, Hamburg, Germany), BioTek ELx405 automatic plate washer (BioTek, Winooski, VT, USA) and BioTek ELx800 plate reader. Individual titers of IBD antibodies were calculated for each sample and used to express the mean geometric titer (Gmean), coefficient of variation (CV) and the number of positive and negative samples for a given flock.

### 2.3. Detection and Differentiation of IBDV by Real Time RT-PCR Evaluation

After collection, BF samples for PCR were suspended in physiological fluid and partially homogenized using glass beads. Total viral genetic material from the prepared samples was isolated using a commercial MagMAX™ CORE Nucleic Acid Purification Kit (Thermo Fisher Scientific, Cambridge, UK) and a KingFisher™ Duo Prime Purification System (Thermo Fisher Scientific, Waltham, MA, USA). The resulting product was used to detect the presence of RNA IBD viruses by Real Time RT-PCR. Real Time RT-PCR analysis was performed using commercial kits (Kylt, Mühlenstraße, Germany) designed for the detection and differentiation of IBD (non-virulent and very virulent strains) using specific primers. Real Time RT-PCR reactions were performed according to the protocol provided by the manufacturer using a BioRad 7500 CFX96 Real Time System thermocycler (BioRad, Boulder, CO, USA). Fluorescence results were read using the dedicated CFX Manager Dx software 3.1 (BioRad, USA).

### 2.4. Histopathology

During necropsy, samples of bursa of Fabricius were fixed in 10% neutralized formalin and later embedded in paraffin blocks. Haematoxylin and eosin staining was used on 5 μm sections. The histopathological lesion scores (HLSs) of the BF samples were evaluated with the Muskett lesion score system [16]. Briefly, when bursa follicles showed no lymphoid depletion and no other lesions were diagnosed, a 0 score was set. When 1–25% of the follicles showed lymphoid depletion with heterophils infiltration, a score of 1 was set. A score of 2 was established when 26–50% showed lymphoid depletion with heterophils infiltration and the necrosis of lymphocytes. A score of 3 was established when 51–75% of follicles showed lymphoid depletion with severe heterophils infiltration and the necrosis of lymphocytes. When nearly complete lymphoid depletion was observed (75–100%) with severe heterophils infiltration, necrosis of lymphocytes and cyst formation, a score of 4 was established. Total lymphoid depletion (100%) with fibrosis and atrophy of bursa gave a score of 5. Individual scores for BF samples were used to calculate the mean HLS for the flock.

### 2.5. Clinical Observations and European Production Efficiency Factor (EPEF)

The European production efficiency factor was calculated with the use of the following formula: EPEF = (survival rate (%) × final body weight (kg)) / (age at slaughter × feed conversion ratio (FCR) (kg/kg)) × 100(1)

An EPEF of 360 was used as a cut-off value. Production cycles with an EPEF below 360 were not considered for further evaluation and the data from those production cycles were discarded.

### 2.6. Statistical Analysis

The results of serological and histopathological examination were analyzed statistically with one-way ANOVA followed by Bonferroni’s post hoc comparisons (Statistica 13.1., StatSoft). Differences were found statistically different if *p* < 0.05. Statistical analysis was performed using individual farm results for each vaccination protocol.

## 3. Results

### 3.1. Clinical Observations and EPEF

In three cases, the EPEF value was below 360. In those cases, additional factors influencing the health conditions of the birds were identified based on standard diagnostic procedures: two cases were identified as an aMPV infection and one as an outbreak of Marek’s disease. In addition, there were two cases where the presence of vvIBDV was detected. All of these five cases were discarded from further analysis. 

The mean EPEF value for the rest of the tested broiler flocks reached 411.6. In none of these flocks did the vet report any clinical symptoms which could be associated with IBD. 

### 3.2. Laboratory Results

The results of the serological evaluations of day-old chicks (MDAs levels) and the age of the birds at the time of first (vaccination protocol I–III) and second vaccination (vaccination protocol II and III), as well as the results of HLS evaluations 1 and 2 and the serological evaluation at the end of the production cycle, are summarized in Table 1. None of the flocks tested had a CV > 40% when the chicks’ serum was tested for the Deventer formula. Birds on farms vaccinated with protocol II were characterized by the lowest antibody titers in both ELISA tests used. At the same time, at week 6, these birds had the lowest HLSs in the BF. The highest IDEXX titers were obtained in birds vaccinated with protocol III, while in the BIOCHEK test the highest titers were obtained for birds vaccinated with protocol I. 

Protocols I and III were characterized by very similar mean HLSs in the BF samples at 6 weeks (Table 1). Differences were found also with regard to the degree of post-vaccination immune uniformity (CV%) (the highest CV% was recorded for vaccination protocol II) and with regard to the number of positive samples in the ELISA test (the lowest number was recorded for vaccination protocol II).

In two flocks, the presence of field IBD virus genetic material was confirmed at 6 weeks of age (one positive result each for a flock vaccinated with protocol I and II, respectively). These flocks were discarded from further analysis. The presence of vaccine virus genetic material was detected in two flocks vaccinated with protocol II.

## 4. Discussion

Gumboro disease is one of the most serious epidemiological and economic problems in large-scale chicken production. The disease is found worldwide and causes both direct losses (morbidity, mortality and increased condemnation) and indirect losses resulting from immunosuppression. The above justifies the regular use of specific prophylaxis in poultry flocks against the disease [1,2,3,4,9,17,18,19].

There are two basic strategies for immunizing chickens against IBD. One is the immunization of reproductive flocks using live and inactivated vaccines to stimulate the production and transmission of MDAs to eggs and the chicks that hatch from them. These antibodies protect maturing B lymphocytes from IBDV infection [1,2,10,12,20]. From the results of our own study, it appears that the flocks included in our study were well protected by MDAs against IBDV. The geometric mean titers that were obtained for flocks covered by each vaccination protocol ranged from 4697 to 5190. It is additionally noteworthy that in flocks vaccinated with protocol I (single vaccination with the intermediate plus vaccine), the average day that this vaccination occurred was at 14.85 days of the birds’ life. These results correspond with our previous observations [21] indicating that there were high levels of MDAs in day-old chicks. 

The second strategy for immunizing chickens is to actively immunize them through vaccination [1,3,12,13,16,20]. Vaccines used for immunization of chicken broilers can most broadly be divided into live attenuated vaccines and new-generation vaccines (vector and immunocomplex vaccines). Despite the advantages of new-generation vaccines, one of the most widely used vaccination protocols involves vaccination with live attenuated vaccines administered with drinking water at the optimal time determined by the Deventer formula. One of the disadvantages of live attenuated vaccines is that they retain some degree of the pathogenic properties of B lymphocytes, and their use alone can lead to severe damage to the structure of the BF [6,12,16,22,23,24]. For the interpretation of HLSs in the BF under field conditions, it is very important to take into consideration the dynamics of virus replication and spread within the flock. While under laboratory conditions (all of the birds vaccinated at the same time and/or the SPF birds were used), the peak of HLS should occur at the same time in all of the birds and it is estimated at 7–14 days after vaccination [25]. Additionally, the mean HLS recorded in laboratory vaccination studies is usually much lower than the one recorded under the field conditions. Under the field conditions it should be taken into consideration the fact of mass-vaccination with drinking water, residues of MDAs at the time of vaccination and vaccine spreading in the flock. Therefore, in such an experiment the peak of HLS in the BF is suspected later at 2–4 weeks after the vaccination [25,26]. Reviewing the literature in this aspect, it should be noted that the degree of HLS in the BF samples recorded in our study were very similar to the degree of lesions found by other authors under laboratory and/or field conditions using IBD vaccines of similar MDAs breakthroughs [16,22,24]. For example, it was estimated that the average HLS for hot vaccines (MDAs breakthrough = 636) at week 4 after vaccination of birds, under laboratory conditions, was between 3.75 and 4.5 [24]. On the other hand, Olesen et al. [25] considered HLSs of 4 or more, under field conditions, for vaccinated flocks (with an intermediate vaccine, MDAs breakthrough = 250) to be considered as suspected of field infection. The above data, as well as the mean HLSs recorded in flocks with three different IBD vaccination protocols, indicate that all of the vaccination protocols used in our study can be considered as safe with regard to the level of damage caused in the BF 2–4 weeks after the vaccination. What is additionally interesting is that in our study, the average HLSs in BF samples did not exceed those considered as indicators of field virus breakthrough, despite the fact that the vast majority of herds analyzed in our study were vaccinated twice against IBD. The above indicates once again the safety of the specific prophylaxis programs described and tested under this study.

Comparing all three vaccination protocols, a clear relationship can be seen between the frequency of vaccination and the type of vaccine, on one hand, and the degree of stimulation of anti-IBDV specific antibody production on the other. The highest antibody titers (regardless of the ELISA used) were recorded in flocks vaccinated once or twice with the intermediate plus vaccine. IBD vaccines can cause damage to the B lymphocytes in the BF. Under field conditions (*per os* vaccination, in the presence of MDAs), the highest degree of such damage is recorded approximately 2–4 weeks after vaccination [16,22,23,24]. From the results of our own studies, it appears that the immunogenicity of vaccines correlates directly with their potential to induce lesions in the BF. The higher the titer of MDAs a given vaccine is able to break through, the more it tends to induce greater HLSs. On the other hand, such vaccines stimulate higher immune responses for protection against IBD. 

## 5. Conclusions

In conclusion, the three tested vaccination protocols were estimated as safe (based on the degree of BF lesions) and effective in the context of protection against field virus infection. The estimated values of serological baseline and the degree of HP lesions in the BF presents a very clear picture of the differences between the different vaccination protocols and allows their adaptation for different farms, depending on the current epidemiological situation of IBD. In the light of the above, in cases of higher field IBDV pressure, more invasive (and therefore more immunogenic) vaccines and vaccination protocols should be used. In the case of lower field pressure, less invasive vaccination protocols should be considered.

## Figures and Tables

**Table 1 animals-14-03116-t001:** Results of MDAs level evaluation of day-old chicks, vaccination timing (mean birds age in days for Vaccination 1 and Vaccination 2), results of histopathological lesion score (HLS) and serological evaluation of three different IBD vaccination protocols.

**Vaccination protocol I**	**MDAs level**	**Vaccination 1**	**Vaccination. 2**	**HLS1**	**IDEXX**	**CV**	**Pos**	**Neg**	**BIOCHEK**	**CV**	**Pos**	**Neg**	**HLS2**
Vet practice 1 *	6139.00	16.00	-	-	3665.00	34.10	23.00	0.00	12,014.00	8.90	23.00	0.00	4.40
Vet practice 2	4996.00	14.75	-	-	3708.00	22.85	23.00	0.00	12,605.00	7.13	23.00	0.00	4.25
Vet practice 3	5140.86	14.86	-	-	3172.14	27.49	23.00	0.00	12,888.00	11.97	23.00	0.00	3.74
Mean values **	5086.69	14.85	-	-	3503.85 ^b^	26.18 ^a^	23,00	0.00	12,977.77 ^a^	10.18 ^a^	23.00	0.00	3.95 ^a^
**Vaccination protocol II**	**MDAs level**	**Vaccination 1**	**Vaccination. 2**	**HLS1**	**IDEXX**	**CV**	**Pos**	**Neg**	**BIOCHEK**	**CV**	**Pos**	**Neg**	**HLS2**
Vet practice 1	6260.50	17.00	22.00	0.30	232.00	119.25	6.00	17.00	1183.00	141.55	15.50	7.50	1.90
Vet practice 2	5026.20	16.20	21.20	0.30	3063.00	27.57	23.00	0.00	14,735.33	7.00	23.00	0.00	nd ****
Vet practice 3	4558.00	16.63	21.63	0.58	2519.63	42.00	20.13	2.88	9239.88	32.94	20.88	2.13	3.40
Mean values	5190.57	16.93	21.93	0.46 ^a^***	2293.08 ^c^	50.55 ^b^	18.62	4.38	9268.54 ^b^	43.66 ^b^	20.54	2.46	3.10 ^b^
**Vaccination protocol III**	**MDAs level**	**Vaccination 1**	**Vaccination. 2**	**HLS1**	**IDEXX**	**CV**	**Pos**	**Neg**	**BIOCHEK**	**CV**	**Pos**	**Neg**	**HLS2**
Vet practice 1	3720	12.00	17.00	1	4296	26	23	0	11,997	10	23	0	Vet practice 1
Vet practice 2	6618.50	15.00	20.00	0.70	4729.00	31.45	23.00	0.00	15,324.00	12.00	23.00	0.00	Vet practice 2
Vet practice 3	4611.38	13.25	18.25	0.23	4051.50	22.64	23.00	0.00	11,686.00	9.98	23.00	0.00	Vet practice 3
Vet practice 4	3974.00	13.00	18.00	0.13	3992.00	26.50	23.00	0.00	12,578.00	11.67	23.00	0.00	Vet practice 4
Mean values	4697.86	13.36	18.36	0.33 ^a^	4153.00 ^a^	24.99 ^a^	23.00	0.00	12,419.07 ^a^	10.61 ^a^	23.00	0.00	Mean values

* results presented for particular vet practice represent mean values of the results obtained for farms under the supervision of a given vet practice; ** mean values were calculated based of the results of individual farms, independently of a vet practice, within the vaccination protocol; *** a–c—mean value with a different subscription letter represent statistical difference, for given parameter, between different vaccination protocols. Statistical analysis was performed using individual farm results for each vaccination protocol independently of a vet practice; **** nd—not done

## Data Availability

The original contributions presented in this study are included in the article; further inquiries can be directed to the corresponding author.

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
