# Peer review of "Comparison of Three Different IBD Vaccination Protocols in Broiler Chicken—Efficacy, Serological Baselines and Histo-Pathological Lesions in the Bursa of Fabricius"

_animals, 2024, doi:10.3390/ani14213116_

Round 1
Reviewer 1 Report
Comments and Suggestions for Authors
In the manuscript entitled “Comparison of three different IBD vaccination protocols in broiler chickens - efficacy, serological baselines and histopathological lesions in the bursa of Fabricii” the Authors compared three strategies for immunizing broilers against infectious bursal disease (IBD) differing in the degree of invasiveness of vaccine viruses. The three different IBD vaccination protocols in broiler chickens, in terms of their effectiveness, estimation of serological baseline using two different ELISA tests, and the degree of progression of BF lesions after vaccination were assessed. The research methods have been properly selected to obtain reliable results. The conclusions drawn by the authors are generally supported by the research results obtained (detailed comments are provided below).
Specific comments:
Line 78: The acronym “MDA” should be explained on first use.
Line 170: The acronym “FCR” should be explained on first use.
Line 175: In cases where one-way ANOVA results were positive, did the authors use post-hoc analysis for multiple comparisons to show differences between individual pairs of groups? If so, what test was used?
Line 275: The Authors indicate that “the immunogenicity of vaccines correlates directly with their potential to induce lesions in the BF”. Did the Authors calculate the correlation coefficient for these parameters?
Line 277: There should be probably “HLS” instead of “BLS”.
Line 315: There should be “Avian Dis” (capital letter).
Author Response
Dear Editor and Reviewers
We are extremely grateful to You for taking the time to evaluate our scientific work. In addition, we would like to thank You sincerely for all Your comments, which undoubtedly contributed to improving the quality of our work. We would like to respond to Your comments and feedback below.
Specific comments:
Line 78: The acronym “MDA” should be explained on first use.
Response: Thank You for this comment. We have made appropriate modifications in the manuscript body.
Line 170: The acronym “FCR” should be explained on first use.
Response: Thank You for this comment. We have made appropriate modifications in the manuscript body.
Line 175: In cases where one-way ANOVA results were positive, did the authors use post-hoc analysis for multiple comparisons to show differences between individual pairs of groups? If so, what test was used?
Response: This is an honest mistake in the manuscript body, from our statistician. We have made appropriate modifications in the “statistical analysis” section.
Line 275: The Authors indicate that “the immunogenicity of vaccines correlates directly with their potential to induce lesions in the BF”. Did the Authors calculate the correlation coefficient for these parameters?
Response: No we did not calculate the correlation coefficient for these parameters. This conclusion was drawn based on the observation of trends in our results.
Line 277: There should be probably “HLS” instead of “BLS”.
Response: Yes, You are definitely right. This was just a typo and it has been corrected.
Line 315: There should be “Avian Dis” (capital letter).
Response: Yes. Once again a typo. Corrected.
Reviewer 2 Report
Comments and Suggestions for Authors
Although the authors are attempting to make comparisons among the three different vaccination protocol, it lacks clarity in the main text. I am providing some:
1. In Abstract: There are some terms required elaboration for the first time use
2. Need to improve the introduction section with more concise rationale why this study is required supported by appropriate references.
3. The reader expects to know the Deventer formula that you have used many times without appropriate description and reference
4. Line 103-106: Not clear—adequate means what? Clearly describe the process you have followed, mentioning the dispenser, intake, and farming conditions as well.
5. Line 108-109: What is the intermediate plus vaccine and why Avi Shield IBD Plus is Intermediate Plus? What is MDA?
6. Line 111-117: Need to write in more details
7. Line 119-126: Who has developed the protocol and how? Sample collection process.
8. Line 131-139: Not clear - need to incorporate appropriate references and revise this with clear description
9. Line 142: How did you collect? Did you follow ethical culling of the birds? Where is the protocol? Need a clear description of the sample collection process - no?
10. Line 168-171: Who calculated this—the author or someone else. Where are the references?
11. Line 180: What is the unit?
12. Vaccination schedule is not mentioned
13. Need to provide the primer sequence
14. The author mentioned about the measurement of production performance, but I could not find it in the manuscript how they did that.
15. The discussion section should be more robust, centering your main finding - no?
Need to improve
Author Response
Dear Editor and Reviewers
I am extremely grateful to You for taking the time to evaluate our scientific work. In addition, I would like to thank You sincerely for all Your comments, which undoubtedly contributed to improving the quality of our work. I would like to respond to Your comments and feedback below.
Specific comments:
- In Abstract: There are some terms required elaboration for the first time use
Response. Thank You for this comment. We’ve made corrections in order to meet reviewers criteria.
Need to improve the introduction section with more concise rationale why this study is required supported by appropriate references.
Response: Thank You very much for this comment. We’ve made some modifications in the section „Introduction – Aim of the study” in order to meet reviewers criteria.
The reader expects to know the Deventer formula that you have used many times without appropriate description and reference
Response: The Deventer formula has already beeen described briefly in the Introduction of the paper. We have made some modifications in order for the desrciption to be more comprehensive.
Line 103-106: Not clear—adequate means what? Clearly describe the process you have followed, mentioning the dispenser, intake, and farming conditions as well.
Response: Thank You for this comment. We’ve made some corrections in section 2.1. in order to meet reviewers criteria.
Line 108-109: What is the intermediate plus vaccine and why Avi Shield IBD Plus is Intermediate Plus? What is MDA?
Response: Thank You for this comment. Both definitions (of intermediate plus vaccines and MDA) are explained in the Introduction of the paper. Avi Shield Plus is an intermediate plus vaccine beause the producer of the vaccine have tested it and registered this product as an intermediate plus vaccine, which represent the level of MDA which the vaccine can overcome at the time of birds vaccination.
Line 111-117: Need to write in more details- Line 131-139: Not clear - need to incorporate appropriate references and revise this with clear description
Response to comments 6 and 8: The details of the Deventer formula has been described in the introduction section. As authors of the article we see no need for unnecessary repetitions.
Line 119-126: Who has developed the protocol and how? Sample collection process.
Response: This is our own protocol and it has been developed based on the previously published papers in order to make our results suitablde to be compared with other data.
9. Line 142: How did you collect? Did you follow ethical culling of the birds? Where is the protocol? Need a clear description of the sample collection process - no?
Response: Thank You for this comment. The data requested by the reviewer was already presented in the section „Institutional Review Board Statement”, while the protocol of the sample collection is described in details in section 2.1 „Experimental layout”.
Line 168-171: Who calculated this—the author or someone else. Where are the references?
Response: This is our own calculations and the reults are presented in table 1.
Line 180: What is the unit?
Response: The is no unit for EPEF. This is just a numerical information about the production performance.
Vaccination schedule is not mentioned
Response: We are very sorry, but we don’t know what does it reffers to. Unfortunately, the reviewer didn’t indicate the lines in the manuscript body for this comment.
Need to provide the primer sequence
Response: As mentioned in the paper commercial PCR kits were used in the study. Therefore the sequences of the primers are confidential. The results of the wor can be reproduced by any other institution with the use of the same PCR kits, which are described and listed in details in the manuscript.
The author mentioned about the measurement of production performance, but I could not find it in the manuscript how they did that.
Response: In our paper the production results were calculated based on the EPEF value (see section 2.5 and 3.1 for details). We also used the cut-off value of EPEF = 360 in order to discard the flocks with the value below 360, which could have been associated with a disease in the flock. Discarding the data from these flocks allowed to increase the reliability of the final results presented in the paper.
The discussion section should be more robust, centering your main finding - no?
Response: As authors, we would like to point out that the results have been centralized in a single table attached to the article, which in our opinion makes them very transparent. We have made some corrections in the "Discussion" section adequate to the changes made in the "Aim of the study" section. In the discussion, the results have been compared with the work of other researchers, which makes this section more interesting and adds more to this area of scientific knowledge. The results of the work are once again summarized in the last paragraph of the Discussion section and in the final conclusions to the article.
Comments on the Quality of English Language
Need to improve –
Response: The paper was corrected by the Native Speaker in order to improve the quality of the English Language.
Round 2
Reviewer 2 Report
Comments and Suggestions for Authors
Can be considered for publishing in the journal